# Sinking Behavior of Netting Panels Made with Various Twine Materials, Solidity Ratios, Knot Types, and Leadline Weights in Flume Tank

Chenxu Shan [1], Hao Tang [1,2,3,4,5,*] , Nyatchouba Nsangue Bruno Thierry [1,*], Wei Liu [6], Feng Zhang [1], Meixi Zhu [1], Can Zhang [1], Liuxiong Xu [1,2,3,4,5] and Fuxiang Hu [7]

1   College of Marine Sciences, Shanghai Ocean University, Shanghai 201306, China; 17551861292@139.com (C.S.); zf10384@163.com (F.Z.); zhumx2021@163.com (M.Z.); canzh@foxmail.com (C.Z.); lxxu@shou.edu.cn (L.X.)
2   National Engineering Research Center for Oceanic Fisheries, Shanghai 201306, China
3   Key Laboratory of Oceanic Fisheries Exploration, Ministry of Agriculture and Rural Affairs, Shanghai 201306, China
4   The Key Laboratory of Sustainable Exploitation of Oceanic Fisheries Resources, Shanghai Ocean University, Ministry of Education, Shanghai 201306, China
5   Scientific Observing and Experimental Station of Oceanic Fishery Resources, Ministry of Agriculture and Rural Affairs, Shanghai 201306, China
6   Key Laboratory of Oceanic and Polar Fisheries, Ministry of Agriculture and Rural Affairs, East China Sea Fisheries Research Institute, Chinese Academy of Fishery Sciences, Shanghai 200090, China; 18817771740@163.com
7   Faculty of Marine Science, Tokyo University of Marine Science and Technology, Minato, Tokyo 108-8477, Japan; fuxiang@kaiyodai.ac.jp
*   Correspondence: htang@shou.edu.cn (H.T.); nnbt@shou.edu.cn (N.N.B.T.); Tel.: +86-21-61900309 (H.T) +86-186-21829173 (N.N.B.T.); Fax: +86-21-61900304 (H.T.)

**Abstract:** Netting is an important component of fishing gear design, and its ability to sink determines the effectiveness of fishing gears such as purse seines, falling nets, and stick-held nets. Therefore, it is crucial to thoroughly investigate the sinking parameters (sinking depth and sinking speed) of the netting panel as a function of the leadline weights using various twine materials, knot types, and solidity ratios. In this study, a generalized additive model (GAM) was utilized to analyze the impact of each factor on the sinking performances of the netting. The results revealed that the sinking depth of the netting was positively correlated with sinking time and leadline weight. However, the netting featured a maximum sinking depth limit, indicating that the sinking depth would not increase beyond a leadline weight of 69.5 g. During the initial phase of the sinking process, the sinking velocity of each netting panel initially increased but gradually decreased over time. The incorporation of a leadline weight reduced sinking time. Thereby, polyester netting exhibited the shortest average sinking time. A comparison of netting types with similar solidity ratios showed that the maximum sinking depth of the nylon netting was 13.20% and 10.11% greater than that of polyethylene and polyester nettings, respectively. In addition, nylon nets' time average sinking speed was 64.58% and 4.62% greater than that of polyethylene and polyester nettings, respectively. The analysis of the GAM model clearly showed that the leadline weight has a significant effect on the netting sinking speed and depth. To ensure that the netting can reach its maximum sinking speed, it is strongly recommended to use nylon and polyester nettings with a low solidity ratio, i.e., a lower twine diameter and greater mesh size with a higher leadline weight, when constructing fishing gear such as purse seines with higher net leadline weights.

**Keywords:** netting; sinking speed; leadline weight; solidity ratio; flume tank experiment

## 1. Introduction

Netting is a crucial component of fishing gear design that is particularly important for minimizing the hydrodynamic forces of fishing gear and maximizing its sinking parameters.

Therefore, knowledge of the variations of the sinking characteristics of nettings is crucial for the study of the effectiveness of some fishing gear, such as purse seines, falling nets, and stick-held nets. This includes catch performance, high-quality fishing, increasing regulations, and global environmental awareness, which are the main drivers of many improvements in the fishing gear research and development taking place worldwide [1–3]. Many researchers have studied the sinking behavior of fishing gear, including netting panels, and identified that the key factors for increasing free school fishing efficiency in the tuna and small pelagic fisheries were the net sinking speed and sinking depth, which can lead to decreased fish escape [1,2]. Thus, the netting characteristics such as solidity ratio (mesh size, twine diameter, and mesh opening angle), twine materials, and knot types, in addition to the leadline weight, are considered to be the key factors affecting the sinking performance of the fishing gear, which in turn affects the success rate of capture [3,4].

Throughout the last few decades, many studies have used physical model testing and numerical simulations to explore the sinking behavior of fishing gear. Thus, increasing the leadline weight was found to be the key to enhanced sinking performance. Feng [5] studied the sinking behavior of nettings with different leadline weights and found that a higher leadline weight increased sinking speed. Xu et al. [6] demonstrated that the leadline weight contributed to the fishing netting sinking performance. However, the other elements that can influence the sinking behavior of the netting panel are the netting design parameters [7]. As an example, Misund et al. [8] and Widagdo et al. [9] demonstrated that the use of larger mesh in the purse seine could improve the sinking performance. Kawamoto et al. [10] found that substituting 30 sections of 300 mm mesh for 28 sections of 240 mm mesh increased the sinking rate of seven percent.

Moreover, modifying fishing gear has recently become a popular approach for reducing bycatch and improving the sustainability of many fisheries. Previously, researchers explored the effect of mesh size, netting material, and hanging ratio on fishing net performance. Thus, Kim and Park [11] found that the use of higher-density material such as netting twine material increased the sinking performance. Feng [12] claimed that for the same leadline weight, the hexagonal mesh could achieve lower drag and greater sinking speed than the square and diamond meshes. This is because hexagonal meshes have a greater mesh opening angle than diamond meshes and are subject to less water drag. Thus, some studies have found that the sinking performances of the purse seine using hexagonal meshes were greater than those of diamond meshes [13].

The performance of the purse sinking and netting features would be considerably affected by sea conditions, setting patterns, and flow velocities, among other variables. Several studies on the impact of various variables on the sinking performance of purse seines have revealed that, in addition to changing the qualities of the fishing gear itself, factors such as the sea's environmental variables must also be considered [14]. Thereby, Wang [15] found that the current flow was inversely associated with sinking performance. Li et al. [16] found that the current speed was positively correlated with the sinking speed of the tuna purse seine using flume tank measurements. In addition, Zhou et al. [17] conducted multiple regression statistical analysis to investigate the factors that influence fishing gear sinking performance and found that current speed has a substantial impact on fishing gear sinking performance. Tang et al. [1,4] discovered that the length-to-height ratio was the most important gear parameter for purse seines' ability to sink based on an analysis of the collected sea data and the correlation between the sinking characteristics of tuna purse seines with operating parameters and operation conditions. In contrast, the key environmental determinants were the current speed and direction. Li et al. [18] used multiple regression statistical analysis to examine how leadline weight affected the sinking performance of the falling net, and they found that as leadline weight increased, the average values of both sinking depth and sinking speed increased. Yan et al. [19] used a generalized additive model (GAM) to investigate the relationship between the sinking performance and catch rate of falling nets and found a positive correlation between catch rate and sinking performance.

Previous studies have primarily focused on the sinking performances of nettings made of a particular material and the factors that influence them. However, various twine material combinations are commonly utilized, and the change in netting materials results in variable sinking performance attributes of different parts of the gear. Therefore, it is critical to investigate the differences in the sinking performances of nettings made of various twine materials, assess the comprehensive effects of many factors on the sinking performance of netting, and thoroughly comprehend the changes in the overall sinking performance of fishing gear. The main purpose of this study is to experimentally analyze the effects of solidity ratios, twine materials, and knot types on the sinking performance of netting panels under various leadline weights. To record the sinking process, gap-type hydrometers and cameras were used, and the generalized additive model (GAM) model was employed to analyze the relative influence of each component affecting the netting's ability to sink and to provide the theoretical foundation for doing so. The outcomes of optimizing fishing gear design parameters and fishing operational aspects can be used to predict and improve sinking performance.

## 2. Material and Methods

### 2.1. Experiment Samples

In this study, nine (9) knotless nettings and one (1) knotted netting often used in designing aquaculture cages and tuna purse seines in China and Japan were investigated. All the nettings were built using nylon (PA), polyethylene (PE), and polyester (PES) twine materials with diamond-shaped meshes (Nichimo Co., Osaka, Japan). Their structural parameters are shown in Table 1. Nylon, polyester, and polyethylene netting materials had densities of 1.14 g/cm$^3$, 1.38 g/cm$^3$, and 0.94 g/cm$^3$, respectively. Mesh size and twine diameters were measured using a pair of digital callipers with a resolution of 0.01 mm at ten (10) different spots on the netting panels. The structural parameter measurements were performed in wet conditions. The solidity ratio is the ratio between the projected area of a netting panel and its outline area, i.e., the total area enclosed by the netting panel. According to Thierry et al. [20], the solidity ratio of a netting panel is expressed as follows:

$$\alpha = \frac{d(2l \pm d)}{l^2 \cos 2\varphi} \tag{1}$$

where $d$ is the twine diameter, $l$ is the bar length, $\varphi$ is the mesh opening angle, "+" represents knotted netting, and "$-$" represents knotless netting. The mesh opening angle was maintained at 45° during the experiments.

**Table 1.** Structure parameters of the netting used in this study arranged according to their solidity ratio.

| Netting No. | Twine Materials | Knot Types | Twine Diameter (mm) | Mesh Length (mm) | Mesh Number (T × N) | Solidity Ratio |
|---|---|---|---|---|---|---|
| net-1 | nylon | knotless | 2.046 | 90 | 7 × 11 | 0.044 |
| net-2 | nylon | knotless | 1.976 | 100 | 15 × 24 | 0.077 |
| net-3 | nylon | knotless | 1.852 | 90 | 15 × 24 | 0.080 |
| net-4 | nylon | knotted | 2.17 | 90 | 15 × 24 | 0.094 |
| net-5 | nylon | knotless | 3.688 | 90 | 15 × 17 | 0.157 |
| net-6 | nylon | knotless | 4.461 | 90 | 15 × 18 | 0.188 |
| net-7 | polyethylene | knotless | 3.205 | 120 | 11 × 18 | 0.104 |
| net-8 | polyethylene | knotless | 3.591 | 75 | 17 × 29 | 0.182 |
| net-9 | polyethylene | knotless | 4.087 | 75 | 17 × 29 | 0.206 |
| net-10 | polyester | knotless | 2.087 | 180 | 7 × 11 | 0.045 |
| net-11 | polyester | knotless | 2.046 | 90 | 15 × 22 | 0.088 |

### 2.2. Experimental Process

Experiments were conducted in the flume tank at Tokyo University of Marine Sciences and Technology (9.0 m in length, 2.2 m in width, and 1.6 m in depth). This flume tank is a horizontal and circular flume tank where the flow is driven by four contra-rotating impellers using constant-speed hydraulic delivery pumps with impellers 1.6 m in diameter, delivering a flow speed ranging from 0.1 to 2.0 m/s. The netting was cut before the experiment to fit the assembly scale of the net panel frame size. The dimension of all the test netting samples was 1.5 m in in length and 1 m in width. The top part of the netting was attached to a cylindrical wooden rod that measured 1.5 m in length, 15 mm in diameter, and weighed 96.82 g. A steel bar measuring 1 m in length, 2 mm in diameter, and 24.5 g in weight was used to secure the bottom part of the netting. The nylon wire is attached to the netting on both sides, and the overall length was kept at 1.5 m. The depth of the steel bar was proportional to that of the pressure value created by the tiny pressure sensor (P306A-03, SSK, Osaka, Japan), which was mounted in the center of the steel bar. The depth data were created by converting the voltage signal after it had been sent to the strain gauge, computer, and signal converter (Figure 1). During the experiment, the water density was 999.9 kg/m$^3$, and the water temperature remained at 17.6–18.4 °C.

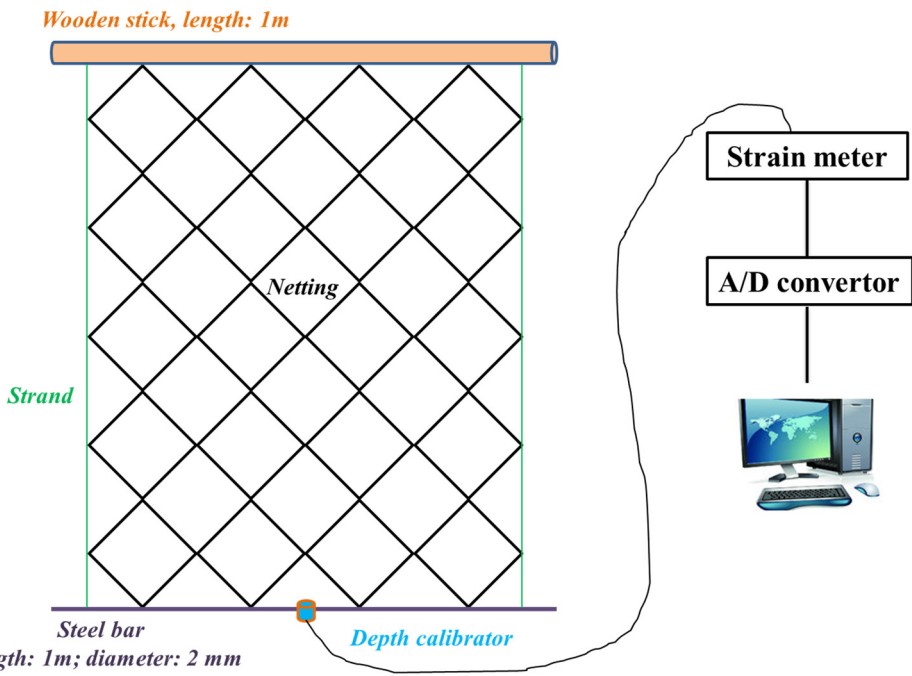

**Figure 1.** Schematic diagram of experimental apparatus for the experimental depth measurements of netting panels.

The assembled netting-frame system was piled on the netting platform to maintain similarity with the settling procedure of the physical model test. The netting platform was a foam-built planar construction in the horizontal plane. Two rings joined the wooden rod at the top of the netting frame, ensuring it remained horizontal. First, the netting was released, and then, by sliding the release platform, the netting that had been set up on the platform began to sink. A camera mounted in front of the flume tank was used to capture the entire sinking process, and the data from the pressure sensors were compared and adjusted for each other (Figure 2). To examine the relationship between the bottom frame weight and the sinking behavior, each netting panel was tested with a total of five (5) leadline weights of 15 g, 30 g, 45 g, 60 g, and 75 g on the original steel bar weight of 24.5 g fixed at the bottom part of each netting. The counterweight of the sinker outline was a 1 m-long chain that weighed 15 g/m on average. In order to increase leadline weights, the chains were uniformly linked with nylon netting at the lower outline of the steel bar after

each series of experiments. Each set of experiments was performed 3 times, for a total of 198 times, in order to prevent experimental mistakes.

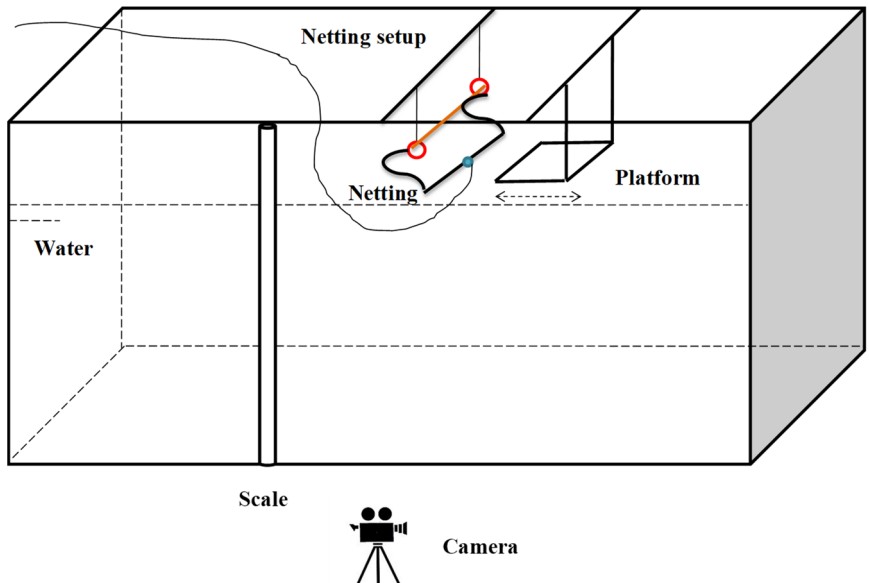

**Figure 2.** Schematic diagram of setup and apparatus for setting on experimental netting.

### 2.3. Data Analysis

During the experimental test, the data were captured at 4 Hz, and 80 data points were recorded in 20 s with a time interval of 0.25 s between different data. Before the experiment, the micro-pressure sensor was set to determine the voltage signal formed by the pressure in the water depth as a function of depth. A digital camera was placed at the front view of the flume tank observation port to record the netting panel behavior in order to describe the sinking process (time and depth). The physical tests were carried out without and with the sensor in order to determine the effect of the pressure sensor on the nets' sinking parameters. Thereby, it was demonstrated by the recorded video that the pressure sensor did not affect the netting sinking parameters.

The sinking process was divided into three (3) phases to better characterize the experimental results: the early phase, the middle phase, and the later phase of the sinking process. The net was buried in water until it achieved its maximum sinking speed in the early phase; it was between the early and late phases in the middle phase; and it reached its maximum sinking depth at a sinking speed of zero in the late phase (Figure 3).

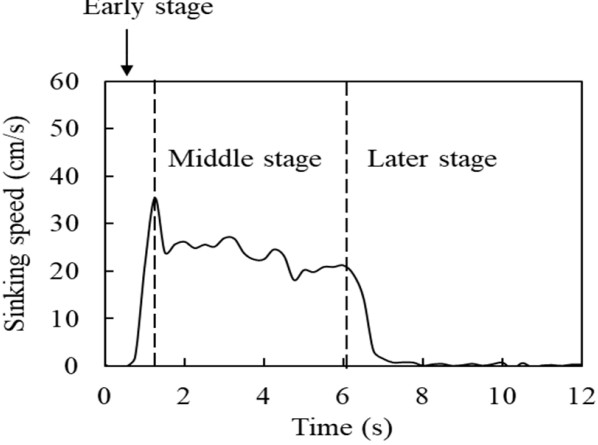

**Figure 3.** Three stages of the sinking process.

In this study, a Shapiro–Wilk test was used to assess the normalcy of sinking depth and sinking speed. To evaluate the effects of various factors such as twine materials, solidity ratios, leadline weights, and knot types on the sinking behavior of the netting panel (sinking speed and sinking depth) and ascertain the influence of various elements on sinking performance, generalized additive models (GAMs) with an identity link function and Gaussian error distribution were used. GAMs are regression models in which the linear predictor depends on unknown smoothing functions of some predictor variables instead of linear coefficients as covariates [21,22]. The GAMs can be expressed as follows:

$$D = s(T) + s(W) + s(\alpha) + M + K + \varepsilon \tag{2}$$

$$Sp = s(T) + s(W) + s(\alpha) + s(M) + s(K) + \varepsilon \tag{3}$$

where $D$ is the maximum sinking depth of the netting panel; $Sp$ is the time average sinking speed; $T$ is the sinking time; $\alpha$ is the solidity ratio; $W$ is the sinking weight (counterweight); $K$ is the knot type, including the knotted and knotless; and $M$ is the twine materials, including nylon, polyethylene and polyester. Function $s()$ are one-dimensional smooth functions of variables. $K$ and $M$ are the categorical variables. $\varepsilon$ is the residual error subject to normal distribution ($F(\varepsilon) = 0$, $\varepsilon = \sigma^2$).

The maximum degree of freedom for univariate terms was chosen at 7 (k = 7) in order to avoid model overfitting. The model selection criteria were chi-square statistical significance tests and Akaike information criterion (AIC). To determine the best model, a stepwise backward selection process was used.

## 3. Results

### 3.1. Effect of Leadline Weights, Twine Materials, and Solidity Ratio on the Sinking Depth

The sinking depth increased as the leadline weight increased, varying from 2.5% to 13.33%. It also increased at the sinking time t < 10 s, while at t > 10 s the sinking depth tended to be constant. For the PA netting, the sinking depth increased as the net solidity ratio increased from 0.044 to 0.094. In contrast, at the solidity ratio greater than 0.094, the sinking depth decreased as the solidity ratio increased. However, for the PE netting, the sinking depth decreased as the solidity ratio increased and increased as the solidity ratio increased for PES netting. The leadline weight was observed to be the critical factor affecting sinking depth. On average, the sinking depth obtained at the leadline weight of 24.5 g was about 4.5%, 46.7%, and 3.6% lower than that obtained at the leadline weight of 99.5 g for the PA, PE, and PES nettings, respectively (Figure 4).

### 3.2. Effect of Leadline Weights, Twine Materials, and Solidity Ratios on the Sinking Speed

Figure 5 shows that the sinking speed of the PES netting was 7.14% and 26.19% greater than that of the PA and PE nettings, respectively (Figure 5). In terms of sinking time, the PA netting clearly performed substantially better than the PE netting. Throughout the middle stage of the sinking operation process, the sinking speed of the PE netting changed significantly and was slower than that of the PA and PES nettings. The PES netting had a lower maximum sinking speed than the PA and PE nettings by 33.3% and 42.9%, respectively. Each netting's sinking speed virtually peaked at a sinking time of 2 s.

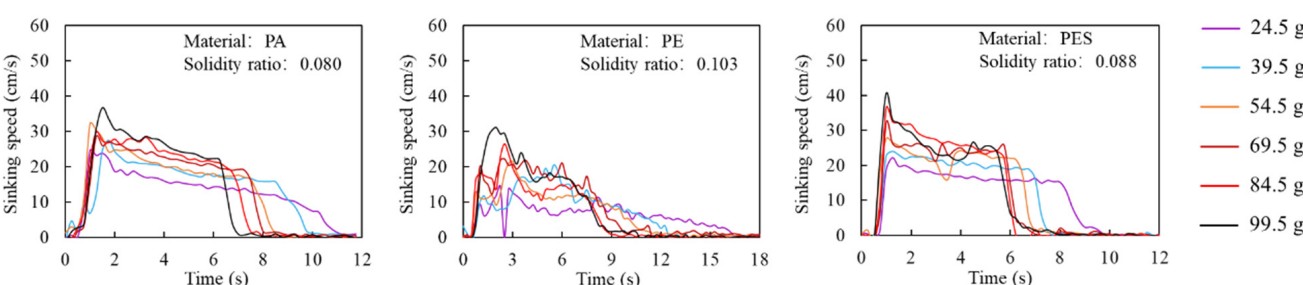

**Figure 4.** Temporal evolution of the sinking depth.

**Figure 5.** The relationships between sinking speed and time of three material nettings ((**left**): PA; (**middle**): PE; (**right**): PES).

### 3.3. Effect of Knot Types, Solidity Ratios, and Leadline Weights on the Sinking Speed of PA Netting

Figure 6 shows that the sinking speed of the knotless nylon nets is greater than that of the knotted nylon nets, with a variation from 6.25% to 21.05%. In the early stage of the sinking process operation, the sinking speed of the nylon netting changed dramatically and increased rapidly due to the leadline weight. Due to the greater netting motions, this

sinking speed achieved a maximum at the sinking time of 2 s; then, due to the interaction between the water and netting, this sinking speed started to progressively reduce at sinking times greater than 2 s and reached 0 at t > 7 s. At the lower leadline weight (24.5 g), the maximum sinking speed of 32 cm/s was attained at the net solidity ratio of 0.188, while at the higher leadline weight (99.5 g), this maximum sinking speed was attained at the net solidity ratios of 0.044 and 0.080. For instance, a leadline weight of 99.5 g reduced the average sinking time by 41.9% and increased the maximum sinking speed by 29.5% when compared to a leadline weight of 24.5 g. The sinking speed of the nylon netting slowed as the solidity ratio increased.

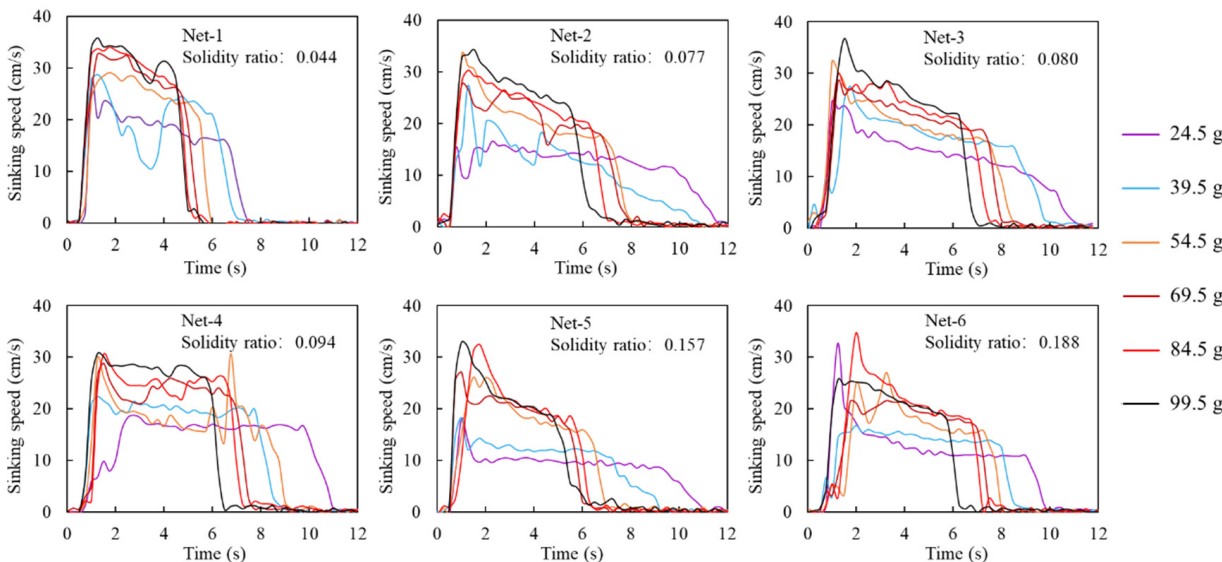

**Figure 6.** The relationships between sinking speed of nylon netting with different leadline weights and sinking time.

### 3.4. Effect of Solidity Ratios and Leadline Weights on the Sinking Speed of PE Netting

The sinking speed of the PE netting increased with a decreasing solidity ratio, ranging from 6.45% to 12.90% (Figure 7). This sinking speed reached its maximum after 3 s and increased as the leadline weight increased. At t > 3 s, the sinking speed decreased as the sinking time increased, with a decreasing rate of 2.31 cm/s. On average, the sinking speed of the PE netting was 20 cm/s at the leadline weight of 24.5 g; it was 59.3% lower than the average sinking speed obtained at 99.5 g (See Figure 7).

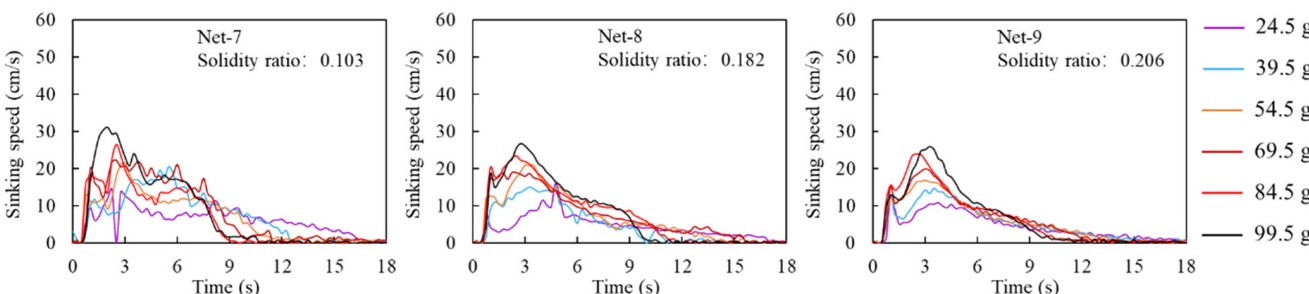

**Figure 7.** The relationships between sinking speed of polyethylene netting with different leadline weights and sinking times.

### 3.5. Variation of Sinking Speed of Polyester Netting

For all solidity ratios, PES netting obtained a maximum sinking speed (40 cm/s) at a sinking time of 2 s (Figure 8). The sinking speed of PES nets increased as the solidity ratio increased, with a variance of less than 5%. At t > 2 s, the sinking speed of the PES

nettings declined at all leadline weights at a rate of 2.10 cm/s. Furthermore, the sinking time required for the sinking speed to attain a steady state is 13.3–32.5% shorter when compared to the base leadline weight. PES netting sinks faster than PA and PE nettings, while PES netting with a lower solidity ratio sinks faster on average than PES netting with a higher solidity ratio.

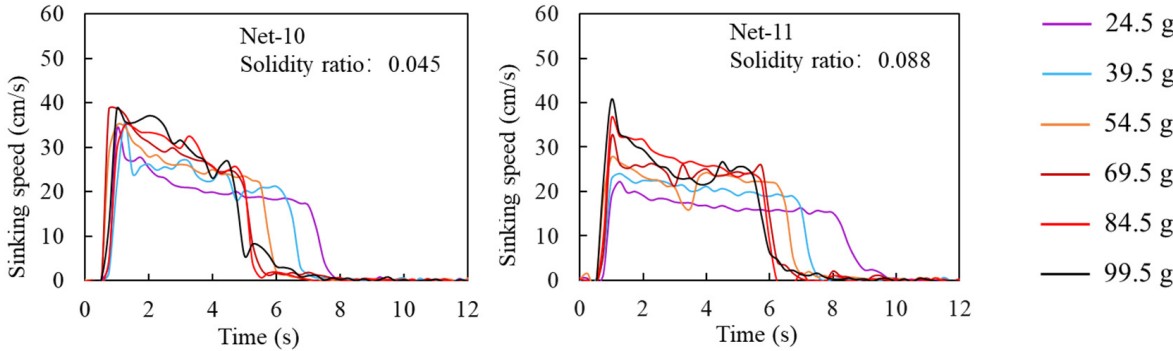

**Figure 8.** The relationships between sinking speed of polyester netting with different leadline weights and sinking time.

### 3.6. Characterization of Netting's Sinking by GAM Model

The GAM results indicated that the effects of twine materials, knot types, leadline weights, and solidity ratios were significant in the sinking speed. The solidity ratio was observed to have a nonlinear relationship with the sinking speed (EDF = 2.695), while the leadline weight had a linear relationship with the average sinking speed (EDF = 1). The GAM model findings suggest that leadline weight, twine materiel, and knot type had the greatest influence on sinking speed (Table 2).

**Table 2.** Summary results of the GAM.

| Variable | Estimate | Standard Error | t-Value | Pr (>\|t\|) | AIC |
|---|---|---|---|---|---|
| (Intercept) | 17.1226 | 0.3208 | 55.378 | $<2 \times 10^{-16}$ | 264.7436 |
| Factor(material) PA Knotted | 3.0475 | 0.8162 | 3.734 | $4.3 \times 10^{-4}$ | 277.5755 |
| Factor(material) PE | −6.7938 | 0.6005 | −11.313 | $2.44 \times 10^{-16}$ | 350.6972 |
| Factor(material) PES | 1.8184 | 0.6164 | 2.950 | $4.57 \times 10^{-3}$ | 350.6972 |
| **Variable** | **EDF** | **Ref.df** | **F-Value** | ***p*-Value** | **AIC** |
| Solidity ratio | 2.695 | 3.143 | 39.35 | $<2 \times 10^{-16}$ | 336.7810 |
| Leadline weight | 1.000 | 1.000 | 272.75 | $<2 \times 10^{-16}$ | 376.7792 |

Note: The AIC value for this model as the designated factor was eliminated. With all factors reserved, AIC is equal to 769.224; deviance explained = 94.2%; R-sq.(adj) = 0.936; *p*-values are presented for all significant (*p* < 0.05) single terms; EDF, estimated degrees of freedom; Ref. df, reference degrees of freedom.

The GAM model revealed a relationship between the average sinking speed and other parameters (Figure 9). The sinking speed decreases as the solidity ratio increases. According to the GAM results, the leadline weight had a linear relationship with sinking speed, as we mentioned above. Furthermore, the sinking speed of PA and PES nettings was greater than that of PE netting. The GAM analysis demonstrated the fluctuation pattern of the influence of several parameters on sinking speed.

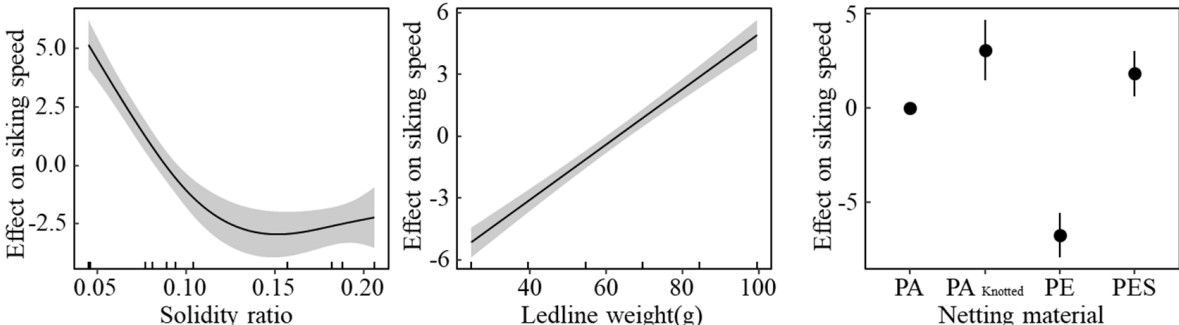

**Figure 9.** Estimated smoothness of three variables on the average sinking speed of experimental netting; y-axes are the partial effect of the variable; shadow sections are the standard error confidence intervals.

As shown in Table 3, the nylon and polyester nettings have better sinking behavior than the polyethylene nets due to their higher density. It was found that the more considerable sinking speed was obtained at the lower solidity ratio (0.044 and 0.045). Thus, decreasing the twine diameter and increasing the mesh size results in higher sinking speed and depth (Table 3).

**Table 3.** Mean values of the sinking speed and sinking depth for each netting from GAM results.

| Netting No. | Twine Material | Solidity Ratio | Sinking Speed | Sinking Depth |
|---|---|---|---|---|
| net-1 | nylon | 0.044 | $24.00 \pm 4.97$ | $123.70 \pm 4.48$ |
| net-2 | nylon | 0.077 | $16.58 \pm 3.57$ | $144.61 \pm 5.98$ |
| net-3 | nylon | 0.080 | $18.41 \pm 4.76$ | $154.28 \pm 3.47$ |
| net-4 | nylon | 0.094 | $19.63 \pm 3.40$ | $148.23 \pm 0.75$ |
| net-5 | nylon | 0.157 | $12.84 \pm 4.73$ | $107.23 \pm 5.19$ |
| net-6 | nylon | 0.188 | $15.66 \pm 3.83$ | $116.48 \pm 6.83$ |
| net-7 | polyethylene | 0.104 | $8.79 \pm 4.44$ | $133.92 \pm 19.3$ |
| net-8 | polyethylene | 0.182 | $8.65 \pm 4.12$ | $111.87 \pm 23.47$ |
| net-9 | polyethylene | 0.206 | $8.01 \pm 2.62$ | $109.02 \pm 9.91$ |
| net-10 | polyester | 0.045 | $22.89 \pm 2.25$ | $134.89 \pm 3.25$ |
| net-11 | polyester | 0.088 | $19.98 \pm 3.90$ | $138.67 \pm 7.29$ |

## 4. Discussion

### 4.1. Influence of the Leadline Weight on the Netting Sinking Behavior

The leadline weight is an important gear component that determines how fishing gears such as net cages and purse seines operate. This parameter influences the sinking performance of the netting structure used in purse seine fisheries throughout fishing operations, such as the sinking depth and sinking speed, as well as the time required to set the net, such as sinking time [1,2]. Thus, the present study and that of Tang et al. [1] demonstrated that the increasing leadline weight led to an increasing sinking depth and speed and a decreasing sinking time. Also, relevant studies such as Dong et al. [23] in the flume tank demonstrated that the bottom weight (sinking depth) increased the sinking depth of the net cage, which agrees with the results obtained in the present study. Thus, it can be observed that in the case of purse seines, the improvement of the sinking performance of this fishing gear (more incredible sinking speed and depth) is achieved when the leadline weight (sinking depth) is higher. In the case of the net cage, the greater sinking depth and volume can also be achieved at the greater bottom weight (sinking depth). It was noteworthy that Hosseini et al. [3] also found a maximum sinking speed of 0.31 m/s and a sinking depth of about 153 m at the higher sinking weight for the Korean tuna purse seine. Li et al. [16] found that the leadline weight significantly improved the sinking performance. Thus, because sinking parameters were the primary performance characteristics for the fishing net in this study, it was shown that the rationalized assembly of leadline weight is critical and should

be considered while examining the law of sinking behavior. According to the studies of Kim [24], it was found that no matter what twine materials are used for the netting design, the sinking parameters of the netting greatly depend on the leadline weight. Despite the fact that the leadline weight is proportional to the sinking speed, consistently increasing the leadline weight has an influence on it. However, it was established that increasing the leadline weight excessively does not considerably increase the sinking speed of the netting. In this study, Cui et al. [25] demonstrated in this investigation that the sinking speed of the leadline is proportional to the half time of the leadline load, and that the same weight load has a substantial influence on the sinking of the wing terminal through sea trails. Hence, the relationship between the average sinking speed of the leadline and the sinking force can be expressed by a simple theoretical calculation as: $S_p = 0.36\sqrt{(f + 0.6f_n)/D}$, where $S_p$ is the time average sinking speed, $D$ is the sinking depth, $f$ is the sinking force per unit length of the sinking load, and $f_n$ is the gravity force per unit length of the sinking net clothing in water [26]. Because the above method is based on the field net tension formula, it ignores the water resistance in the netting and the leadline weight. The hydrodynamic coefficients are proportional to the mesh parameters and the Reynolds number. Variable netting mesh parameters and speed motions result in variable hydrodynamic coefficients and hydrodynamic forces. Although Tang et al. [1] found that in natural conditions the operational parameters that most affected the sinking performance of the tuna purse seine were current speed and direction, it is important to note that in order to target more pelagic species, a higher leadline weight is required.

### 4.2. Effect of Solidity Ratio on the Netting Sinking Behavior

On the other hand, it was confirmed that d/l did not fully describe the mesh geometry properties and could not characterize shrinkage, so the solidity ratio is an essential factor that can be used to characterize the mesh geometry more comprehensively [27]. However, the present study, on the other hand, demonstrated that the sinking speed and solidity ratio had a nonlinear relationship. Furthermore, because of the netting density, this investigation demonstrated that the netting weight is the other element determining the sinking speed variation. Thus, according to Tang et al. [4] and Zhou et al. [28], the maximum sinking depth of the purse seine was obtained at the higher mesh size. That agreed with the results described in this study, which certified that the sinking depth of PA and PE nettings increased as the mesh size increased. However, unlike the previous studies of Tang et al. [4] and Zhou et al. [28], and the results obtained on the PA and PE netting panels, it was observed that the sinking depth of the PES netting panels decreased as the mesh size decreased. The reason for this was that the sinking performance of the netting panel or purse seine is determined by the mesh size and is heavily influenced by the twine diameter. As a result, the analysis of this study revealed that the sinking depth increased as the twine diameter decreased, allowing us to conclude that the solidity ratio had a significant influence on the sinking behavior of the netting panel. This study also discovered that, for the same leadline weight and solidity ratio, knotted netting had a greater sinking performance compared to knotless netting. This was because the twine length of the knotted netting was greater than that of knotless netting, resulting in the greater weight of knotted netting.

The solidity ratio was positively related to the drag coefficient for the netting panel perpendicular to the flow, whereas the opposite result was obtained when the netting panel was placed parallel to the flow [1]. As the netting was folded during the experiment, the netting panel sank quickly under the pull of the guideline and had a greater attack angle with the flow direction in the early stage. Thereby, the attack angle slightly depends on the netting sinking behavior. Therefore, it is not possible to measure the influence of the solidity ratio on the netting drag solely on the basis of its effect on the sinking speed. From a scientific view, it is possible to measure the relative motion of the netting in the water flow at different stages of sinking. Based on this measurement, the net drag at the different stages can be determined.

*4.3. Effect of Twine Materials and Knot Types on the Netting Sinking Behavior*

This study showed that other factors, such as the knot types and the twine materials, can also influence the netting sinking performance. Thus, it was found that the PES and nylon netting's sinking performance was higher than the PE netting's sinking performance. This was in agreement with the purse seine used in the Japanese tuna purse seine fisheries, which mainly utilizes knotless polyester. Also, the results of the present study agreed with those obtained by Kim [24], which demonstrated that the average sinking speed of the Korean purse seine was more significant when the purse seine model was designed using PES and PA twine materials. Indeed, the results of the present study showed that the sinking speed of PES netting was in the range of 20–30 cm/s, while the sinking speeds of PA and PE nettings were in the range of 15–25 cm/s and 5–15 cm/s, respectively. That demonstrated that because the PES and PA have a higher material density, their sinking speed was greater compared to that of PE nettings. Thus, it can be concluded that the use of the PES and PA nettings have great importance with regard to the improvement of sinking performance compared to PE netting material. Moreover, by using netting panels with the identical mesh sizes made of PA and PE netting materials, Hosseini et al. [3] showed that polyester's sinking behavior behaved better than nylon's using a numerical simulation method. That confirmed the results we obtained in the present study. Additionally, another reason, apart from what we mentioned above, was that because of the higher resistance performance of PE netting material, the PE nettings have a lower sinking performance compared to PA and PES nettings. The other reason was that, according to Thierry et al. [20], the hydrodynamic forces of PA and PES netting were lower than those obtained using PE netting, which is probably why the nettings made with PES and PA nettings sink faster and have better sinking performance compared to the ones made with PE netting under the same other conditions. Thus, according to Kim et al. [29] and the present study, a greater-density model net sinks faster than a lower-density model net. The present results were also in agreement with those obtained during the sea trial by Nomura [30], who demonstrated that the purse seine made of nylon and polyester material sank faster than that of Cremona material. The less pliable the twine material, the stiffer it is. However, the softer materials of nylon and polyester have better sinking characteristics. Apart from that, knotless netting has better sinking performance than knotted netting because knotted netting has higher water resistance than knotless netting.

**5. Conclusions**

The effect of solidity ratio, leadline weight, knot type, and twine material on the sinking behavior of the netting panel used in purse seine fisheries was experimentally investigated. A GAM model was used to analyze the sinking depth and speed under different leadline weights, net gear design parameters, and operational methods. The main conclusions of the study are presented below:

(1) It was found that increasing sinking time resulted in increasing sinking depth. Meanwhile, the increased leadline weight helped to significantly enhance sinking behavior.

(2) The sinking performances of several netting twine materials differed significantly, and the solidity ratio and sinking speed were negatively correlated. Polyester nettings had the best sinking performance as compared to the other netting materials, and we recommended the using of PES and PA materials for developing purse seines to obtain better sinking performance and thus a higher catch performance. Furthermore, it was established that knotted netting sinks less efficiently than knotless netting.

(3) Our results can be used to improve and optimize the design of fishing gear. It is recommended to use polyester twine material, a lower solidity ratio (lower twine diameter and greater mesh size), and a higher leadline weight with the values beyond 69.5 g corresponding to the leadline of 13 kg/m used in real conditions by Chinese tuna purse seine.

**Author Contributions:** Conceptualization, C.S., H.T., N.N.B.T. and F.H.; methodology, C.S., H.T. and N.N.B.T.; software, C.S. and H.T.; validation, C.S., H.T. and N.N.B.T.; formal analysis, C.S., H.T. and F.H.; investigation, C.S., H.T., N.N.B.T. and W.L.; resources, C.S., H.T., N.N.B.T., C.Z. and F.Z.; data curation, C.S., H.T., W.L. and M.Z.; writing—original draft, C.S., H.T. and N.N.B.T.; writing—review and editing, H.T., N.N.B.T. and L.X.; visualization, C.S., H.T. and N.N.B.T.; supervision, H.T., N.N.B.T. and L.X.; project administration, H.T. and N.N.B.T.; funding acquisition, H.T. and N.N.B.T. All authors have read and agreed to the published version of the manuscript.

**Funding:** This study was financially sponsored by the National Natural Science Foundation of China (Grand No. 32373187 and 32350410404) and Natural Science Foundation of Shanghai (23ZR1427000).

**Data Availability Statement:** Data will be available upon request by the corresponding authors Hao Tang and Nyatchouba Nsangue Bruno Thierry.

**Conflicts of Interest:** The authors declare no conflict of interest.

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
