# Peer review of "Sinking Behavior of Netting Panels Made with Various Twine Materials, Solidity Ratios, Knot Types, and Leadline Weights in Flume Tank"

_jmse, doi:10.3390/jmse11101972_

Round 1

Reviewer 1 Report

The manuscript (MS) includes an experimental study in microcosm scale according to commercial fishing gear, nets dimension conducted at sea by testing sinking speed with some factors or properties of the nets. 

The introduction of the MS was good enough detailed on the topics of the hypothesis of the present study.  The authors claimed that previous studies study the sinking rate with the limited factors out of the factors involved into the MS. 

Material and methods described how to make the experiment in a tank in small size (miniature model) of the net and environment (tank) compared to sea fishing size. Such multiplication were here so ignored in the assumption regarding the real size of the net at aquatic environment. Some points could be arisen to be clarified if there are effect on the results of the MS,

Tank is too shallow, so in this case, the vertical force of water pressure is ignored. 

Circulation and water movement is homogeneous unlike the condition at sea. Including the water column depth traveled by the nets the 3D effects (horizontal and vertical water forces)  start and water lines (thermocline, halocline etc.) appear through the column when the net is deployed at real fishing area. Water density, and components; temperature, salinity and pressure was ignored

Lead weight is miniature also, and their surface of the area as well. 

All aforementioned affect the sinking speed in reality. 

There are  few more, but as the experiment is designed in a microcosm, the MS is unlikely to simulate the real environment. 

Data analysis needs more applying the rectangular data matrix subjected to proper statistics to optimize the results among the combination of the net factors for the conclusion. 

Solidity equation is not cited. 

Results must be summary in a Table to compare sinking rate among the combination of the net factors.

For the entire MS, too long sentences must be avoided. 

Discussion must compare advantageous and disadvantageous between the experiments and deployment at sea instead of repeated results. A lot repetitions are there as same as the results are.

Conclusion 

led weight always is expected to increase the sinking in water, and an experiment not needed. Tuna purse seine must be not comparable with miniature net at a tank condition. It could be better useful one-to-one reduced size miniature based on size of commercial purse seine. 

There are too long sentences, seeming to be generated by a machine. It is really hard to follow. There are grammatical errors, not completed sentences.

In the text, citation must be in [ ], and no dot. 

Among the all results, the optimized and best net factor combination could be advised, supported by the statistical direct results.  

The MS needs a major revision clarifying points arisen in my review. 

There are too long sentences, seeming to be generated by a machine. It is really hard to follow. There are grammatical errors, not completed sentences.

Reviewer 2 Report

This study deals with the sinking characteristics (sinking depth and sinking speed) of different twine materials, knot types, and solidity ratios used in netting when subjected to varying leadline weights. In addition, a generalized additive model (GAM) was utilized to analyze the impact of each factor on the sinking performance of the netting. This study shows some interesting results, though, in order to be published in the journal, major revision must be made as listed below.

[Major items]

(1) The explanation for GAM model in this MS is highly satisfactory in Eq.(2). The "behavior (S)" cannot be explanatory variables mathematically. In addition, why do you use "~" in the equation instead of "="?

(2) Several types of smooth functions can be used in GAM method. PLs elaborate more on this matter in your analysis.

(3) Accuracy of the GAM must be shown in more straightforward manner, instead of just summarizing results in Table 2. The reviewer's strong suggestion is to illustrate the measured value and predicted value by GAM in the same diagram, which will facilitate the direct comparison between them visually.  

(4) The objective GAM application is not clear in this study. The results are valid by fitting for such small-scale experiment, though, it cannot be used for real scale situations with much larger scale of fishing gear.

(5) p.12 Line 328: What the meaning of “Average”? Time averaging? Ensemble averaging? Spatial averaging? Or something else? Define clearly.

(6) p.12 Line 329: The equation for Va is dimensionally incorrect. Sqrt(force/length/length)=speed? In addition, H is sinking depth? However, the notation of “S” has already been used elsewhere? “g" is the sinking force per unit length? If it is so, use another notation, as “g” must conventionally be used for the gravitational acceleration.

[Minor items]

(1)   There cannot be two corresponding authors. Must be a single person from the co-authors.

(2)   The way of citing a paper is not correct. Instead of [1, write as [1].

(3)   Important items are missing at the end of the MS such as "Author Contributions", "Funding", "Data Availability Statement" and "Conflicts of Interest."

PLs check the English thoroughly throughout the MS.

Round 2

Reviewer 1 Report

The manuscript (MS) is currently better than the previous version. However, this ex situ model indeed were performed to guide purse seine fishery. The author optimized the MS by deleting such aim of the MS. This decreases the soundness of the MS. I've just learned that the ex situ water is fresh water. Purse seine fishery is conducted at sea water. Anyway, answering to the comments, most of responses were done in the current revised version.

The MS need needs to overview English of the text with some syntax and grammatical errors, e.g. in line 532, "the net sanks rapidly in ...."

Reviewer 2 Report

The authors have made corrections properly in response to revieweer's comments. It can be now published in the journal.
